# A Toolbox for Construction and Analysis of Speech Datasets

**Evelina Bakhturina**  **Vitaly Lavrukhin**  **Boris Ginsburg**

**NVIDIA, Santa Clara, USA**
{ebakhturina, vlavrukhin, bginsburg}@nvidia.com

## Abstract

Automatic Speech Recognition and Text-to-Speech systems are primarily trained in a supervised fashion and require high-quality, accurately labeled speech datasets. In this work, we examine common problems with speech data and introduce a toolbox for the construction and interactive error analysis of speech datasets. The construction tool is based on Kürzinger et al. work, and, to the best of our knowledge, the dataset exploration tool is the world's first open-source tool of this kind. We demonstrate how to apply these tools to create a Russian speech dataset and analyze existing speech datasets (Multilingual LibriSpeech, Mozilla Common Voice). The tools are open sourced as a part of the NeMo framework.

## 1 Introduction

Speech data can be applied to a wide variety of tasks. The most common tasks include automatic speech recognition (ASR), text-to-speech synthesis (TTS), speech enhancement, etc. In this work, we focus on speech data for ASR, although the discussed techniques are also applicable to other speech processing tasks.

Neural Network (NN) based ASR systems have made a huge leap in recent years [5, 24, 21, 16, 18, 7, 17]. One of the key enabling factors for such progress was the release of the LibriSpeech corpus which contains about 1000 hours of English speech [32]. LibriSpeech became the standard benchmark for English ASR [24, 16]. Existing large non-English speech datasets – AISHELL [4, 10], Mozilla Common Voice (MCV) [2], CSS10 [33], and more recently, the Multilingual LibriSpeech (MLS) [35] – have made a significant contribution towards encouraging ASR model development for non-English languages. However, the creation of new speech datasets remains an ongoing problem, not only for low resource languages, but also for English. The reasons for that are the following:

- **Lexicon**. End-to-end ASR models need to be constantly retrained to adapt to a new domain lexicon and language changes [25]. For example, an ASR model trained on the LibriSpeech could have a low word error rate (WER) on classical books but a high WER when transcribing modern technical speech.

- **Acoustic environment**. ASR models are deployed in multiple settings and they should be robust to different room conditions and input audio quality. For example, intelligent home assistants need to differentiate background conversations from users' requests. Thus, the speech training datasets need to contain a variety of samples with noisy room conditions, multi/single speakers recordings, etc.

- **Speakers variability**. ASR models should be able to transcribe speech from speakers with different backgrounds (geographic origin, accent, gender, age, etc.).

35th Conference on Neural Information Processing Systems (NeurIPS 2021) Track on Datasets and Benchmarks.

The amount of English speech data released in recent years has grown exponentially due to the fact that data, along with model size and compute, remains the main driving force to boost an ASR model's accuracy (see Figure 1). Speech researchers need to analyze the training data in order to understand possible model biases and errors. However, this is becoming increasingly difficult as the size of speech corpora grows. To the best of our knowledge, there is no open tool for interactive exploration and analysis of speech datasets.

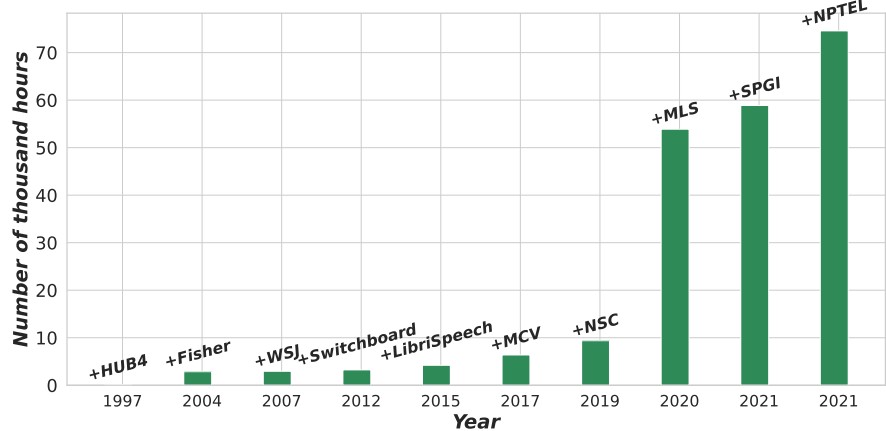

Figure 1: Size of English speech datasets over the years. English Broadcast News Speech Corpora (HUB4) [14], The Fisher Corpus [6], Wall Street Corpus (WSJ) [11, 8], Switchboard [13], Singapore English National Speech Corpus (NSC) [20], SPGISpeech (SPGI) [31], NPTEL2020 Indian English (NPTEL) [29]

The speech data for training ASR models consists of short audio clips (usually up to 20 seconds long) and their text transcriptions. There are two common methods for speech dataset construction: 1) record short text utterances read by speakers, and 2) given a long audio file and its corresponding text, split the audio and text into short utterances using forced alignment techniques.

The latter approach is gaining popularity due to the growing number of hours of YouTube and similar content released under permissive licenses. The most popular solutions for forced alignment – Gentle [30] and Montreal Forced Aligner [27] – are based on Kaldi [34] hybrid ASR models and, therefore, use intermediate phone representations. The main drawback is that these models require phonetic lexicons, and available public lexicons cannot cover even LibriSpeech [32], not to mention non-English languages. Mozilla DS-Align tool [9] is another forced alignment method based on the end-to-end DeepSpeech ASR model [18]. However, this model is far behind recent end-to-end NeMo ASR models (QuartzNet [21], Citrinet [26]) in terms of accuracy and size. Since more accurate ASR models tend to produce more correct alignments, we propose a tool to create speech datasets by extending the CTC-Segmentation approach introduced by Kürzinger et al. [23] with NeMo ASR models. NeMo provides a wide range of pre-trained ASR models[1] and a recipe for fine-tuning English ASR models on the target language data [25].

We have created a toolbox to ease the analysis of existing speech datasets and construction of new speech corpora for new domains and languages. The main contributions of the paper:

1. We introduce an open-sourced NeMo toolbox[2]: the *CTC-Segmentation tool* - an end-to-end pipeline for splitting long audio files into shorter clips based on the CTC-Segmentation algorithm [23] and *Speech Data Explorer (SDE) tool* for interactive audio data analysis.

2. We describe how to construct a speech dataset using the tools and demonstrate that even a weak initial ASR model could be used to perform audio-text alignment to iteratively increase

---

[1]https://docs.nvidia.com/deeplearning/nemo/user-guide/docs/en/main/asr/results.html#automatic-speech-recognition-models

[2]https://github.com/NVIDIA/NeMo/tree/main/tools

the size of the training data and alignment quality. We conduct experiments on Russian data to support the findings.

3. We provide general guidelines on how to analyze and clean audio data with SDE and demonstrate them on MLS [35] and MCV [2].

## 2 NeMo toolbox

In this section we discuss the tools developed to facilitate speech dataset construction and analysis, see Figure 2.

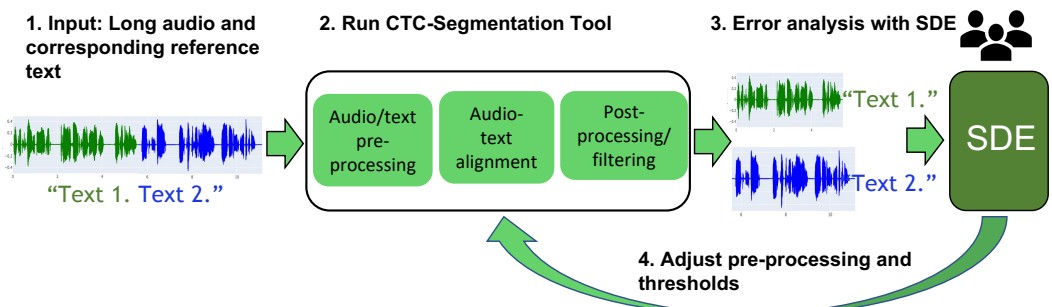

Figure 2: NeMo speech dataset construction/analysis workflow.

### 2.1 CTC-segmentation tool

The CTC-Segmentation algorithm introduced by Kürzinger et al. [23] aligns text utterances within long audio files to prepare short text-audio pairs suitable for ASR model training, i.e., up to 20 seconds long. The algorithm makes use of pre-trained ASR models trained with the Connectionist Temporal Classification (CTC) loss [15] and consists of two stages: 1) The first stage, *forward pass*, is to run inference through a pre-trained ASR model to obtain character probabilities for each time step of the original audio file. 2) The second stage, *backward pass*, determines the most likely path that reconstructs the utterance to be aligned.

During segmentation, the log probability of alignment is calculated for each character. These probabilities are then aggregated into a confidence score for each utterance. The alignment confidence score helps to spot problematic utterances where the transcript and the reference texts may differ, for example, when a reader swaps or skips words.

[23] shows that the CTC-Segmentation alignment method is more robust compared to other tools such as Gentle [30] or Aeneas [1]. Consequently, we decided to integrate the CTC-Segmentation algorithm [23] into an end-to-end NeMo CTC-Segmentation tool for speech dataset construction. Our extension includes the following:

- added support for pre-trained NeMo ASR models and functionality to segment multiple input files in parallel, in order to speed up the segmentation.

- provided audio and text pre- and post-processing scripts, making the tool work end-to-end from the original long text and audio files to the data suitable for ASR model training (see Section 3.1).

- exposed additional parameters to improve segmentation quality, see Section 3.2.

Both the original CTC-Segmentation package[3] and the NeMo CTC-Segmentation tool[4] are released under Apache-2.0 License [36].

---

[3]`https://pypi.org/project/ctc-segmentation/`
[4]`https://colab.research.google.com/github/NVIDIA/NeMo/blob/stable/tutorials/tools/CTC_Segmentation_Tutorial.ipynb`

The CTC-Segmentation tool makes a step towards reducing human involvement in the dataset construction process. With tight enough thresholds on the segmentation confidence score and high-quality ASR models, poorly aligned or normalized segments will be disregarded automatically. The actual amount of human interaction depends on the dataset quality and quantity requirements, the quality of the reference texts, and the ASR model used for segmentation. The larger the number of errors between reference texts and audio and the weaker an ASR model is, the more users need to take control over the segmentation confidence score threshold and dataset analysis (see Section 4).

Section 3 demonstrates the Russian LibriSpeech (RuLS) corpus creation using the NeMo CTC-segmentation tool. Additionally, we discuss the LibriSpeech [32] re-segmentation experiment[5].

## 2.2 Speech data explorer

Speech Data Explorer (SDE) is a web application for interactive exploration and analysis of speech datasets. We designed SDE with the following goals in mind: 1) to get basic statistics on a dataset, e.g., number of hours, number of utterances, alphabet (a set of unique characters), size of vocabulary, histograms of audio file durations and words spoken per second. 2) to explore individual utterances (observe the signal's plot in the time domain, its spectrogram, play audio) in order to develop a basic understanding of the dataset. While the first requirement could be satisfied with a set of scripts, the second one needs an interactive component for data visualization and audio playing. This is why we decided to leverage Plotly Dash[6], a framework for data visualization applications. SDE's interactivity significantly reduces time needed to inspect and analyze speech datasets. Adding transcripts from a pre-trained ASR model and computing a set of error metrics (WER, CER, word accuracy) help to analyze and identify issues either with the dataset or the ASR model.

The SDE features include:

- global dataset statistics (alphabet, vocabulary, duration-based histograms)

- navigation across the dataset using an interactive data-table that supports sorting and filtering

- inspection of individual utterances (plotting waveforms, spectrograms, custom attributes, and playing audio)

- error analysis (word error rate, character error rate, word match rate, word accuracy, display highlighted the difference between the reference text and ASR model prediction)

- audio signal parameters estimation (sample rate, peak level, frequency bandwidth)

Some of the mentioned features are demonstrated in Figure 3. Section 3 describes how to use SDE for error analysis and filtering during speech dataset construction, and Section 4.2 provides a walk-through of an exploration of some existing speech datasets for training acoustic models.

SDE is limited to short ASR utterances (up to few minutes), and longer audio recordings might result in slow response and visualization. SDE was released under Apache 2.0 license as a part of NeMo. To the best of our knowledge, SDE is the first open source tool for interactive exploration and analysis of speech datasets.

## 3 Applications: dataset creation from scratch

This section describes how to create a dataset for training ASR models from long audio and text files using the CTC-Segmentation and SDE tools. A similar approach could be applied to create a TTS dataset [3]. We constructed a Russian dataset for training ASR models from public domain LibriVox audiobooks [7]. The dataset was released previously on OpenSLR[8] platform and is used here to showcase the NeMo tools. The dataset is Public Domain in the USA.

---

[5]`https://docs.nvidia.com/deeplearning/nemo/user-guide/docs/en/v1.5.0/tools/speech_data_explorer.html#sde-demo-instance`

[6]`https://plotly.com/dash/`

[7]`https://librivox.org/`

[8]`https://www.openslr.org/96/`

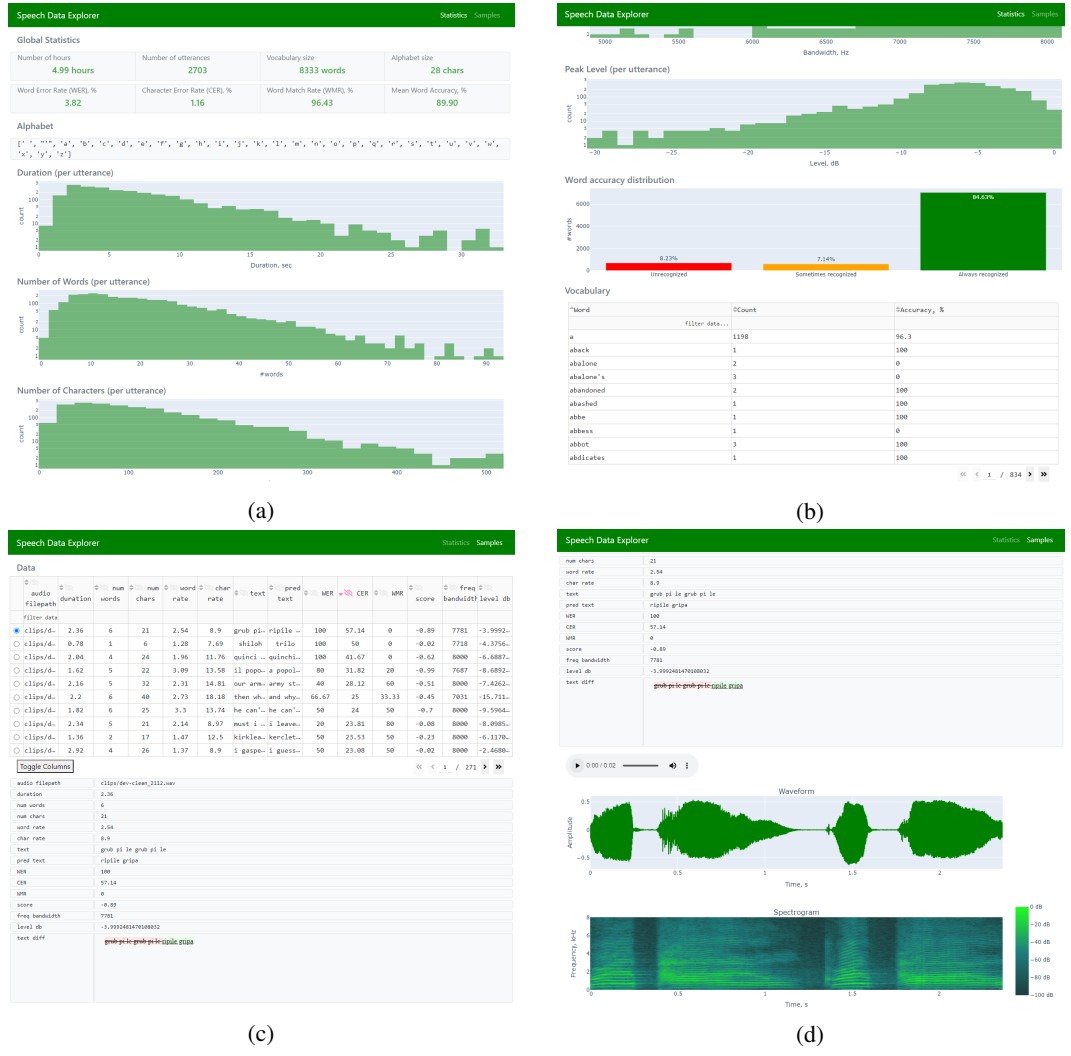

Figure 3: a) SDE Statistics (General), b) SDE Statistics (Vocabulary), c) SDE Samples (Datatable), d) SDE Samples (Player)

## 3.1 Audio and text preprocessing

The audio recordings are directly downloaded from LibriVox.org. We convert original MPEG Layer III (MP3) audio files to the mono Waveform Audio File Format (WAVE) with Linear PCM bitstream sampled at 16 kHz for compatability with NeMo ASR models.

Unlike the English part of the LibriVox project which heavily relies on the Gutenburg project texts[9] in TXT format, the Russian LibriVox section mostly uses reference texts in PDF or JPEG formats. We abandoned the idea of applying a PDF to text converter or using an Optical Character Recognition (OCR) model because 1) the text recovered from book images would contain a significant number of artifacts due to the poor quality of the scanned pages, and 2) many LibriVox Russian books contain archaic orthography which would make OCR perform even worse. Thus, we resort to alternative text resources to obtain TXT versions of the books. The downside of this approach is that the book versions we found could differ from the text used during the audio recording. The mismatch between audio and the reference text could result not only in poor alignments but also hurt the model that uses such data (see Section 3.4 for details).

---

[9]http://gutenberg.org/

Once the reference texts are available, we split long text files into short utterances and prepare the text for alignment. Text pre-processing includes the following steps:

- Normalize numbers. Unfortunately, good normalization tools are rarely open sourced for languages other than English. As a naive solution, we propose a simple conversion of numbers to their digit-based expanded equivalents. For example, '19' is converted to 'one nine' or 'один девять'. This simple normalization procedure does not take audio into account and may lead to a mismatch between the normalized text and what was read by the readers. We exclude the segments with numbers from the final version of the dataset. The aim of this rudimentary normalization step was to make sure the alignments of the subsequent segments are not broken.

- Substitute Russian abbreviations with their full spoken equivalents ('т.д.' -> 'так далее') and normalize non-Russian letters. Many old Russian books contain snippets of French texts, so we apply simple transliteration of non-Russian characters to their closest Russian equivalent ('i' '->' 'и' and 'j' '->' 'ж'). As with the numbers normalization, this step fills the gaps for the successful alignment, and such segments should be dropped.

- Split the original text into short utterances based on the end of sentence punctuation marks. We use regular expressions for this step. In our experiments, we also tried splitting the sentences in the middle to reduce the average duration of the final samples. However, the segmentation results deteriorate with such a strategy, and more partially cut words were observed (see Section 3.4 for details). We believe this is due to the relatively low accuracy of the original Russian ASR model.

## 3.2 Text and audio alignment

After the steps described in Section 3.1, we have a list of normalized text utterances. Next, we need to find the start and end of these utterances within the original long audio clip. During segmentation, we assume that the text utterance order matches the corresponding audio. The CTC-Segmentation requires a pre-trained CTC-based ASR model to extract character log probabilities for each time step. We train a Russian ASR model using cross-language transfer learning [25]. The Russian model was initialized from an English QuartzNet 15x5 [21] [10], which had been trained on 3000 hours of English public speech data. We fine-tune the model on 80 hours of the Russian subset of the MCV dataset (ver. 5.1, licensed under CC-0) using the NovoGrad optimizer [12] with a learning rate of 0.0015, weight decay 0.001, and a cosine annealing learning rate policy. The model was trained for 300K steps with a batch size of 40 per GPU on a single DGX with 8 V100 GPUs. The fine-tuned model has WER of 12.63% on the MCV dev set.

As discussed in Section 3.1, the reference texts might deviate from the audio due to differences between book versions, leading to misaligned audio-text segments. Poorly aligned segments have a low alignment confidence score. To mitigate the issue, we use an alignment score threshold of -2 and disregard all segments with a confidence score lower than the threshold value. Another important parameter is the *minimum window size* that determines the minimum size corresponding to the single utterance inside the audio. In our experiments, we observe that alignment errors could be significantly reduced if we run the aligner multiple times with different window sizes and then select only the matched segments. NeMo segmentation tool uses the following window sizes to produce the final alignments: 8000 (recommended default value), 10000, and 12000 frames.

## 3.3 K-fold re-alignment

The accuracy of the ASR model directly affects the quality of the segmentation. We used a relatively weak ASR model fine-tuned only on a limited amount of speech data (see Section 3.2) to segment the LibriVox recordings. As a result, the WER of the initially segmented corpus was 45.80% vs 12.63% on MCV dev set. Note, some WER degradation is expected as the vocabulary of the LibriVox audiobooks differs from the MCV dataset, which uses more recent texts [2].

To improve the dataset alignment quality, we re-tune the ASR model in a cross-validation manner. Specifically, we create 3 data folds (no book overlap between the folds) and use two folds along with the Russian MCV training set for ASR model fine-tuning and re-segmentation of the third fold. The

---

[10] https://ngc.nvidia.com/catalog/models/nvidia:nemospeechmodels

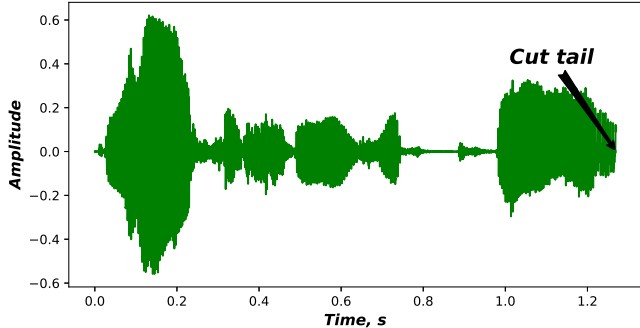

Figure 4: An example of the segmented audio clip waveform with the very last sound of the utterance missing - abrupt ending.

3-fold re-alignment prevents the ASR model from memorizing reference text and alignment errors. It reduces the average book WER from 45.8% to 18.83%.

## 3.4 Error analysis and filtering

We used SDE to evaluate the quality of the final dataset. Overall, the CTC-Segmentation produces relatively accurate alignments given that the time step probabilities were produced by a Russian ASR model fine-tuned on only 80 hours of data and a greedy decoding strategy. We notice that some segmented audio clips end abruptly and miss the ending phonemes of the utterance or include the first phonemes of the next utterance. Figure 4 shows an example, where the audio clip ends before the fading of the waveform occurs. To filter out such examples, we analyzed the mean absolute (MA) signal value at the end of the audio clip in SDE to set a threshold.

To validate the quality of the RuLS dataset, we fine-tune English QuartzNet-15x5 model on both the Russian MCV and the RuLS training sets as described in Section 3.2. As shown in Table 1a, the WER for MCV dev set drops from 12.63% to 9.65% .

Table 1: a) Russian QuartzNet-15x5 model trained on MCV-Ru and combination of MCV-Ru and RuLS. b) Statistics of the RuLS dataset splits.

(a)

| Train set | Greedy WER (MCV, dev set), % |
|---|---|
| MCV | 12.63 |
| MCV + RuLS | 9.65 |

(b)

| Subset | Duration, hours | Number of Speakers |
|---|---|---|
| Train | 92.79 | 5 |
| Dev | 2.81 | 1 |
| Test | 2.65 | 7 |
| Total | 98.25 | 13 |

## 3.5 Dataset splits, statistics, and limitations

The RuLS corpus is complementary to the MLS dataset [35], as Russian language is not present in the MLS. The RuLS is constructed from 17 Russian LibriVox audiobooks, and has 98.2 hours of speech in total. The corpus has three parts: train, dev, and test without speaker overlap (see Table. 1b). The maximum length of audio utterances is 20 seconds (see Figure 5).

The process of the RuLS dataset creation demonstrates the application of the NeMo toolbox. The RuLS dataset has some limitations. It is based on a small number of books due to the lack of existing reference texts. Gender balancing is not considered as most of the thirteen LibriVox volunteers, who participated in the recordings, were male. LibriVox provides metadata to identify a particular book reader, and this information was added to the RuLS dataset to create non-overlapping dataset splits

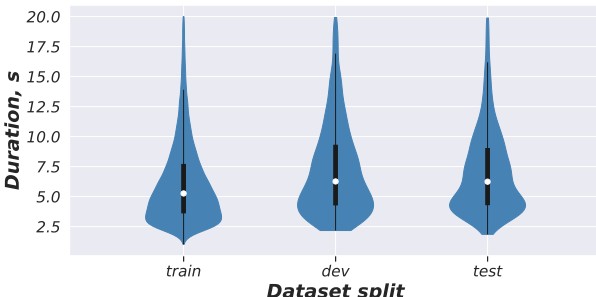

Figure 5: Violin plot of the audio segment lengths for the RuLS dataset splits.

(see Section 5 for the discussion on speech replication ethics). The dataset is based on classic Russian literature that targets different audiences (children, adults), so we do not expect offensive content.

## 4 Applications: speech dataset analysis

### 4.1 General ASR error analysis using SDE

It is not easy to create an error-free speech dataset. Even if native speakers check the utterances manually and validate the data, some mistakes might remain (see Section 4.2). These mistakes could lead to similar errors in transcripts produced by a trained ASR model on new data. For example, in our early experiments with character-based end-to-end convolutional CTC ASR models (like Jasper, QuartzNet) we observed that the models could learn normalization errors from training data. For example, audio "two fifty six" could be transcribed as "two *hundred and* fifty six" . Thus, one needs to explore the speech corpus to understand its errors and biases.

We analyzed many open and commercial speech datasets and noticed the following common issues: 1) *lack of text normalization* - any special character, digit, foreign letter outside of a target alphabet should be either dropped or converted to its spoken form. 2) *segmentation errors*, i.e. the reference text might have extra words that were not spoken, or missing words that were pronounced in the correspondent audio clip, and 3) *completely incorrect reference transcripts*, i.e., no speech in audio or completely different sentence was pronounced.

SDE effectively helps to find utterances with the mentioned issues and to filter speech datasets. If no pre-trained ASR model for the target language or domain is available, then a set of heuristic rules can be used to validate the dataset. Such as: a) check out-of-alphabet characters and out-of-vocabulary words to detect issues with text normalization, b) inspect utterances with high and low character rates (the number of characters in the reference transcript divided by the duration of the audio clip) to spot segmentation errors. Usually, normally paced English speech has average character rate of 15 characters per second. 30 characters per second and higher might indicate extra words in reference transcripts that were not spoken in the audio clip. If a pre-trained ASR model for target language is available, then SDE can import ASR transcripts and compute multiple error metrics for further analysis. The most straightforward filtering rule is to use a threshold on CER to drop inaccurate utterances.

### 4.2 Analysis of existing datasets: MCV, MLS

MCV is a great initiative to construct open speech datasets for multiple languages using a crowd-sourced platform. Volunteers participate in two tasks: recording audio clips for given text sentences, and validating of the recorded utterances (upvoting or downvoting). Each MCV release includes all recorded samples, reference transcripts, and optional metadata provided by volunteers (accent, gender, age range). An audio sample has to get at least two upvotes and more upvotes than downvotes in the validation task to be included in the "validated" subset.

We consider MCV English 7.0 validated subset which has 1975 hours of speech data. Although the dataset is for the English language, its alphabet has 149 characters, see Figure 6. The dataset can be

used neither for training nor for evaluation without text normalization and preprocessing. Therefore, we delete most punctuation characters, replace a single quote character with an apostrophe in cases where it was used as an apostrophe, drop all utterances with foreign letters and special characters that are hard to normalize (e.g., "%" might be pronounced "percent" or "percentage"), see Figure 7.

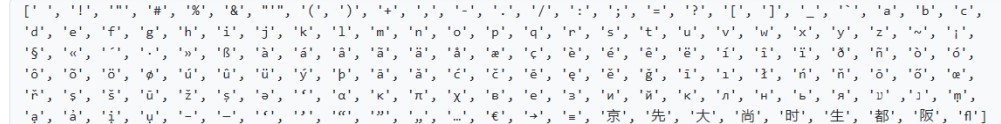

Figure 6: MCV English alphabet

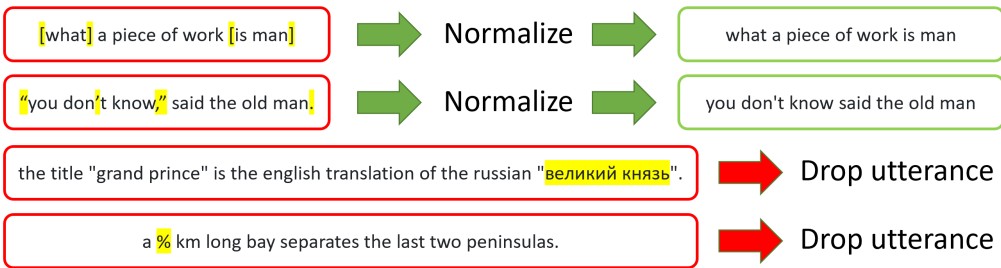

Figure 7: MCV text normalization examples

Even without applying a pre-trained English ASR model, we can observe utterances with a high character rate (up to 127 characters per second). While they have 2 or 3 upvotes and 0 or 1 downvotes, they either contain just noise or incomplete utterances (cut audio due to early stopping of recording). The next step is to add predictions with a pre-trained English ASR model (we use Citrinet-512 from NeMo). Sorting by CER in descending order, we can observe numerous utterances with different issues: audio recordings that do not contain speech at all, repetitions in audio (e.g., a recording containing a reference sentence repeated three times), completely wrong reference texts (speaker pronounced different sentence), non-English reference transcripts (entire German sentence spelled with English letters only). Table 2 shows examples of dropped utterances. While a high CER might be due to an error prone ASR model, the presented examples actually have errors in the reference transcripts. Generally, it is easier to filter the dataset using a threshold on CER than manually re-validate all samples with high CER. After filtering the subset with a CER threshold of 25%, we got 1938 hours, see Table 3 for detailed statistics.

Table 2: Examples of incorrect reference texts in MCV English

| ID | Reference text | ASR transcript | CER, % |
|----|----------------|----------------|--------|
| 1 | six | six i have six plus three grandchildren | 1200 |
| 2 | undefined | but the tradition also says that we should believe the messages of the desert | 778 |
| 3 | beds beaches and lawns are good for reading | [no transcript] | 100 |
| 4 | wer hat diese zeichnung angefertigt | we had de that session and we he it | 66 |
| 5 | although used during the first world war they were sold when the war ended | although used during the f | 65 |

Table 3: Statistics of the MCV English validated subset before and after cleaning

| Subset | Number of utterances | Duration, hours | Alphabet size |
|--------|----------------------|-----------------|---------------|
| Original | 1 425 783 | 1 975 | 149 |
| After text normalization | 1 422 481 | 1 970 | 28 |
| After filtering | 1 401 757 | 1 938 | 28 |

The MLS dataset is the largest speech dataset derived from LibriVox. We analyze the MLS English set. The original train subset contains 44659 hours. Its alphabet includes the standard 28 characters (space,

apostrophe, 26 English letters) and a hyphen symbol. A quick search on words with a hyphen in the SDE vocabulary data table reveals many instances of words like "to-day", "to-morrow", "twenty-five", "it-the", "you-and", etc. So it makes sense to do the following substitutions: "to-" => "to", "-" => " ".

We use a pre-trained Jasper ASR model [24] from NeMo to transcribe the audio. Using SDE we can easily analyse the most frequent words that have zero accuracy (all instances were incorrectly transcribed by the ASR model). This reveals the following issues: lack of text normalization ("mr" => "mister", "mrs" => "missus"), British spelling ("recognise" => "recognize", "realised" => "realized"), missing apostrophe ("dont" => "don't"), missing non-English letters in foreign words ("seor" => "señor", "frulein" => "fräulein", "franois" => "françois"), weird words due to OCR errors ("tlie", "witli", "avhich", "enghsh"). In most cases those errors could be fixed with a small set of substitution rules like "avh" => "wh", "tli" => "th", "httle" => "little", "tiien" => "then", etc. Two books with substantial number of OCR errors were removed completely (IDs: 7756_9499 and 10603_12201). The MLS train set has many segmentation errors (with many extra or missing words in reference transcript). The most straightforward way to filter such errors is to apply a threshold to the CER. After filtering the subset with a CER threshold of 10%, 42967 hours of data remained. The MLS English dev, test sets are more accurate. Still, the dev subset has 6 utterances with Spanish word "señor" (should be dropped). The test subset has 1 utterance containing Russian phrase "активные мероприятия" (should be removed) and 3 utterances with a period character (should be normalized).

Even recent popular English speech datasets like MCV and MLS have issues in reference transcripts that require both additional pre-processing (text normalization) and removal of some utterances (37 hours from MCV and 1692 hours from MLS). SDE is the essential tool to find and analyze issues in speech datasets efficiently and to come up with solutions to fix them.

## 5  Ethics

As most of the ASR models are supervised, they inevitably inherit data biases. It is a good practice to release metadata information along with the dataset for researchers to perform data balancing and bias evaluation. Although many speech dataset creators strive to release gender-balanced data [28] there are many unresolved issues remaining. For instance, [19] shows a profound gap in African American Vernacular English speech recognition due to the lack of the relevant speech data used for training ASR models. There are also some ethical questions, especially for TTS models, as they can potentially be misused. We strongly encourage dataset users to be ethically responsible and get the voice actors' consent before using their voice for synthetic replication.

## 6  Conclusion

Our work focuses on data creation and analysis because 1) more data can improve an ASR model accuracy without changing the underlying architecture, and 2) new target domains and low resource languages require ongoing dataset construction. At the same time, due to the nature of speech data collection, many existing speech datasets have issues that might lead to errors and biases in the trained models. We developed a new toolbox to address the above issues. The toolbox includes the enhanced CTC-based segmentation tool for audio-text alignment and the Speech Data Explorer for interactive error analysis. We demonstrated the efficacy of the toolbox explaining how to construct the Russian LibriSpeech corpus, which improved WER on the MCV Russian *dev* subset from 12.63% to 9.65%, and how to analyze popular English speech datasets MCV and MLS.

The tools and the scripts for cleaning MLS and MCV datasets are open sourced under Apache-2.0 license in the NeMo framework [22] [11].

## Acknowledgments

The authors would like to thank Elena Rastorgueva and Christopher Parisien for their review and feedback. We also would like to express our gratitude to LibriVox volunteers.

---

[11]code: `https://github.com/NVIDIA/NeMo/tree/main/tools`, documentation: `https://docs.nvidia.com/deeplearning/nemo/user-guide/docs/en/main/tools/intro.html`

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
