# OpenReview forum: "A Toolbox for Construction and Analysis of Speech Datasets"
_NeurIPS.cc/2021/Track/Datasets_and_Benchmarks/Round2 — NeurIPS 2021 Datasets and Benchmarks Track (Round 2)_

### Official Review · Reviewer_KEW2 · 2021-09-05
**An incremental contribution building on previous tools for ASR delivered by researchers**

**Rating:** 7
**Confidence:** 3
**Clarity:** 1. The paper is written well overall,…

**Strengths:**

The tools could have an impact on the way speech datasets are created with forced alignment techniques given long audio files and corresponding text. It is likely the code developed will be of high quality and have good compatibility with existing models in the repository, as it was developed under the open-source NeMo framework.

The authors have also demonstrated some useful empirical rules for the creation of datasets by examining different error metrics and common issues, also detailing the steps taken to fix these. As the method integrates ASR models which are near the state of art, this tool could be a really good solution for forced alignment techniques without the presence of phonetic lexicons. There are few other options for this purpose, the closest I could find being DSAlign, which the authors mention.


**Weaknesses:**

There are several weaknesses relating to the two tools introduced, as well as the resulting overall impact and contribution of the paper. I will cover these separately in the following subsections.

### Overall contribution
1. Both the dataset creation and SDE tools are not well linked or visible within the paper. Instead, the URL points to the NeMo main repo, where the tools are not mentioned in the main repo readme and only form a very small submodule of the whole codebase (and I only found them within the tools folder in the repo).

2. It is nice that the tools are well integrated into an existing repo, but it appears the tools will realistically only be useful to those already using NeMo, and thus the potential for future uptake is diminished.

### Dataset creation tool
3. The dataset creation tool itself is strongly built on previous efforts (Kuerzinger's CTC model), with the unique contribution being the compatibility with NeMo pre-trained ASR models, and additional scripts for pre- and post-processing for an end-to-end pipeline. This differs from the expectation set by the abstract, which led me to believe that the methods developed would be more novel. Adjusting the abstract to more accurately reflect the contributions of the paper would alleviate this problem.

### SDE tool

4. The contribution of this tool is relatively minimal. As far as I see the individual utterances example is a simple file print, transcript (and diff if ASR preds are given), an audio play element and a spectrogram.
I haven’t fully understood how SDE was used or is used to find errors in datasets: the paper describes many possibilities of scenarios with a list of heuristics if no pretrained ASR method is available, however I cannot see a clear description of what the tool exactly performs from the paper or GitHub readme. To my understanding, generic statistics are available, as well as the function of stepping through individual examples, from which the users need to make their own decisions on further actions to take.

5. If a pre-trained ASR model for target language is available, then SDE can import ASR transcripts and compute multiple error metrics for further analysis. The most straightforward filtering rule is to use a threshold on CER to drop inaccurate utterances. It is not clear what the role of SDE is in doing this. Does it only compute the metrics and leave the user to choose thresholds and manually filter predictions?

6. The paper does not include a direct link to the tool, and the only screenshot/specific use case is presented in Figure 3.  A user guide with some examples of the visuals of dataset statistics and the decisions made based on those statistics would be of great value here.


**Additional Feedback:**

* If there is a way of hosting the work on an independent repo with more compatibility with other ASR models, uptake could be greatly improved in future. As it stands the tool appears fairly deeply buried within NeMo. This reduces the perceived impact and significance of this work, making it appear to me like two (albeit very well constructed) wrappers for previous methods instead.

* Polishing the figures, better linking the tools with specific examples (e.g. I found a nice tutorial notebook for the creation tool [here](https://github.com/NVIDIA/NeMo/blob/main/tutorials/tools/CTC_Segmentation_Tutorial.ipynb) but no mention of it at all in the paper or appendix!) , and trying to generalise outside of specific applications will be steps that would persuade me to revise my score.

As a final note, please engage constructively with my review during the rebuttal phase, and I would be happy to update my score in accordance with other reviews and your responses. This topic is outside of my strict area of expertise so I acknowledge that I may have made some errors and am receptive to feedback. Thank you!

# Edit: post rebuttal

I have raised my score to 7 following the satisfactory resolution of many of my comments. Thanks to the authors for co-operating and improving their submission based on feedback.

**Correctness:**

See the *Dataset creation tool* section in *Weaknesses*. Beyond this criticism, it appears methods have been implemented well, and the authors have already saved the results of the dataset creation process to an open-sourced library (the Russian LibriSpeech dataset). Design decisions made on the way models were trained on train/dev/test data appear very commendable, helping improve the quality of the final dataset created.

**Documentation:**

Though this paper is about tools and best practices for speech dataset creation, which does not require extensive documentation, the tools themselves are not well linked to the paper, and there is a lack of tutorials and visualisations within the paper to fully understand the benefits of some of the approaches (in particular SDE). Overall I found that I had to put in significant effort to locate the actual tools and scroll through the GitHub repo to find tutorial notebooks and example cases, which others may not be as willing to do.

**Ethics:**

To the best of my knowledge, there are no ethical concerns with this work, all contributions have been appropriately credited. It does however appear that the paragraph on ethics (5.1 Conclusion) has been added on without much thought, and is better suited to another separate section or subheading.

**Relation To Prior Work:**

The paper explains relatively well the differences of its contributions from existing work, though I would like to see a comparison between other forced alignment tools: how does the overall package compare in the production of a dataset? Is the accuracy of the ASR method the only factor in this?

The paper states that
> Since more accurate ASR models tend to produce more correct alignments, ...

Why does this library use the NeMo ASR models in particular when the references given [21, 26] showed that there are other models that perform better ASR? As far as I see the main justification could be the reduction in parameters of the models, but it seems more likely that it is a design decision due to compatibility with methods developed by the author's group. As a result, the tools presented in this paper will mostly only appeal to those already using NeMo (or intending to use NeMo).


**Summary And Contributions:**

This paper introduces open-source tools to construct speech datasets from raw audio alongside transcripts (the NeMo CTC-based end-to-end construction tool). Additionally, a Speech Data Explorer (SDE) is introduced which retrieves statistics and individual utterances of data for exploring existing datasets.

The main contribution of the CTC-based tool (on top of the original CTC method) is to provide audio and text pre- and post-processing scripts, making the tool work end-to-end from the original long text and audio files. The primary application of this tool is for forced alignment of speech and transcripts in the absence of phonetic lexicons.

The authors demonstrate and describe how to create a Russian speech dataset with their tool. Some analysis of existing speech datasets is also performed to indicate a few errors present in existing datasets and showcase the SDE tool.

---

### Official Review · Reviewer_SWAm · 2021-09-16

**Rating:** 6
**Confidence:** 4

**Strengths:**

I've developed several speech datasets in the past (though not involving segmenting longer audios), and there is no open-source tooling for it, so it's great to see work being done on this.

The pre-set dataset visualizations look very useful---I'll give the SDE a try the next time I need to think about the high-level statistics of my dataset.

**Weaknesses:**

I think the tool is a great start, and I hope the authors keep building on it, but it's not yet at the level of quality, originality, or usefulness for a prestigious venue like this.

The main problem for me is it's not clear to me how much of the recipe needs to be done by hand and how much is done by the tool.

e.g. "The CTC-Segmentation requires a pre-trained CTC-based ASR model to extract character log probabilities for each time step. We train a Russian ASR model using cross-language transfer learning" --> Does the toolkit do this for me? If so, what languages are available? If my language is not available, how easy is it for me to supply my own?

So what I'd like to see is: how easy would it be to re-create LibriSpeech using this approach? I'd like to be able to throw the exact same audiobooks that were used to create LibriSpeech into the tool (using a CTC model pre-trained on e.g. Common Voice English) and have LibriSpeech pop out. That would be an extremely convincing demonstration that the tool works well.

I'd love more screenshots of using the tool, e.g. the experiment where utterances are sorted by CER. How easy is it to click on an utterance, listen to it (bring up the screen shown in Figure 3b), and edit the transcript? Suppose that I see "%" in the alphabet generated from the transcripts. Can I open the utterances that contain that weird symbol and listen to them? (Maybe one contains the word "percent" and I can just make that fix by hand.)

Another issue is that, while the Russian LibriSpeech experiment is a nice showcase and definitely shows that re-alignment helps (because you get better WER on Common Voice Russian), it's hard to tell how well the alignment tool works because we don't have the ground truth segmentations. I think a better way to test the quality of the segmentation would be to take a set of transcribed utterances that are already segmented with high confidence (e.g. LibriSpeech), concatenate them into a single audio file, run CTC-Segmentation (and re-training/re-alignment), and see if the algorithm recovers the original segmentation.

Does the tool come with common normalization rules (e.g., "mr" --> "mister", drop utterances with foreign characters)? Or do I have to write them myself?

**Additional Feedback:**

n/a

EDIT: updated score to 6

**Clarity:**

The authors claim to "introduce an open-sourced NeMo toolbox: the CTC-Segmentation tool", but the CTC-Segmentation tool repo they link to is written by Ludwig Kürzinger (not one of the authors). Am I missing something here? Did you re-implement their algorithm, or do you just import their tool?

"NeMo segmentation tool uses the following window sizes to produce the final alignments: 8000 (recommended default value), 10000, and 12000." Is that number seconds, frames?

**Correctness:**

The submission is neither a dataset nor a benchmark; it's a tool for speech data exploration, though I would consider it in-scope for this track.

**Documentation:**

I can't find documentation for the toolkit.

**Relation To Prior Work:**

There is no similar tool for exploration and visualization of speech datasets.

**Summary And Contributions:**

The authors present a tool, the Speech Data Explorer (SDE), that can display useful information about a speech dataset (mean/std duration of utterances, length of transcripts, plot spectrograms/waveforms and play audio).

The authors also present a recipe for creating speech datasets from audiobooks and other very long audio files with transcripts: run a pre-trained CTC ASR model on each audiobook to get label probabilities for each frame, align each audiobook with its transcript using CTC-Segmentation (a simple and robust lexicon-free forced alignment technique introduced in [23]: run Viterbi on the CTC graph but only fill in the dynamic programming table within a small window away from the diagonal, so that the complexity only grows linearly in the length of the audio), then cut the audio at end-of-sentence punctuations to get short utterances that can be used to train a new ASR model. They find that finetuning the model on the transcripts from CTC-Segmentation, and then re-running CTC-Segmentation to get a new set of aligned utterances, yields better WER on the Common Voice test set.

Finally, the authors demonstrate that the SDE can be used to sort utterances by different metrics to remove utterances with obviously incorrect transcripts.

---

> ### Author Response · Authors · 2021-09-26
> **Re: Review Part 1**
>
> Dear Reviewer,
>
> Thank you for your constructive feedback. Below we are addressing each point separately.
>
> > The main problem for me is it's not clear to me how much of the recipe needs to be done by hand and how much is done by the tool.
>
> The actual amount of human involvement during the dataset creation depends on the dataset quality requirements, the quality of the reference texts, and the ASR model used for segmentation. The default thresholds should work out of the box when the reference texts match the audio, and the ASR model has decent accuracy. In that case, human interaction is minimal. However, when reference texts and audio have a significant mismatch or when an ASR model is of poor quality, users are advised to adjust the segmentation confidence score threshold that affects the amount of data retained (i.e., samples with low segmentation confidence scores are removed automatically from the final corpus). Additionally, it is recommended to inspect the final corpus with the SDE tool to validate the quality of the resulting dataset. We adjusted Section 2.1 with a more detailed explanation of the human role in the dataset creation process.
>
> > We train a Russian ASR model using cross-language transfer learning" --> Does the toolkit do this for me? If so, what languages are available? If my language is not available, how easy is it for me to supply my own?
>
> The CTC-Segmentation tool requires a pre-trained CTC-based ASR model. NeMo toolkit provides pre-trained ASR models for English, Spanish, German, Catalan, French, Italian, Polish, and Russian languages. If an ASR model for the target language is unavailable, one needs to train the model from scratch or fine-tune a pre-trained English model for a target language. The dataset creation toolkit is part of the NeMo framework, which provides recipes for training/fine-tuning ASR models. For example, this Colab-based tutorial shows how to take a pre-trained English ASR model and fine-tune it on a target language. During the RuLS dataset creation, we confirmed that even a weak ASR model fine-tuned on a small amount of data could lead to good segmentation results.
>
> > I think a better way to test the quality of the segmentation would be to take a set of transcribed utterances that are already segmented with high confidence (e.g. LibriSpeech)...
>
> Thank you for this idea! We followed your suggestion and re-segmented the LibriSpeech corpus [32] to demonstrate both the CTC-Segmentation and Speech Data Explorer tools. We concatenated all audio files from the dev-clean split into a single file and set up the CTC-Segmentation tool to cut the long audio file into original utterances.  We choose to use only the dev split as the pre-trained ASR model used for segmentation was trained on the LibriSpeech train set. We used the CTC-based QuartzNet15x5Base-En ASR model. The segmented corpus has 3.82% WER and contains 300 out of the initial 323 minutes of audio. The remaining 23 minutes are the silence at the beginning and end of the audio removed during the segmentation. A running instance of the SDE instance demonstrates the re-segmented corpus https://docs.nvidia.com/deeplearning/nemo/user-guide/docs/en/main/tools/speech_data_explorer.html#sde-demo-instance.  Due to the number of pages limit, this experiment is not included in the paper.
>
> Additionally, the CTC Segmentation and SDE tools were used to create a LibriVox-based English dataset for training text-to-speech models ([Hi-Fi TTS dataset](http://www.openslr.org/109/) [3]).
>
> > Does the tool come with common normalization rules (e.g., "mr" --> "mister", drop utterances with foreign characters)? Or do I have to write them myself?
>
> For the English data, we use a comprehensive set of normalization grammars based on Weighted Finite-State Transducers integrated into the toolkit to automatically normalize things like "mr" --> "mister" and “on April 5th” → “on April fifth” automatically. For non-English languages, the tool uses [num2words](https://pypi.org/project/num2words/). Segments with poor normalization or foreign characters have low segmentation scores and are removed from the final corpus automatically if the scores are below the threshold values. Throughout the paper, we use the RuLS corpus example to generalize to the cases when a comprehensive normalization system is not available for the target language.
>
> The remaining answers are posted in a separate comment due to the number of characters limit.
>
> Kind regards,
>
> Paper246 Authors

---

> > ### Comment · Reviewer_SWAm · 2021-09-27
> > **response**
> >
> > Thanks for your response and for running the LibriSpeech experiment---can you clarify what you mean by "the segmented corpus has 3.82% WER"?
> >
> > What I meant when I said you could validate the tool using existing segmentations like LibriSpeech was _"there should be no words in a transcript that are not in the segmented utterance"_---in other words, if I run the tool on an audiobook containing just "first sentence second sentence", and the tool segments the audio into utterances containing "first sentence" and "second sentence", then the transcripts should not be e.g. "first sentence second" and "sentence". Does that make sense?

---

> > > ### Author Response · Authors · 2021-09-28
> > > **Re: response**
> > >
> > > Dear Reviewer,
> > >
> > > Thank you for your fast response!
> > >
> > > > Thanks for your response and for running the LibriSpeech experiment---can you clarify what you mean by "the segmented corpus has 3.82% WER"?
> > > What I meant when I said you could validate the tool using existing segmentations like LibriSpeech was "there should be no words in a transcript that are not in the segmented utterance"---in other words, if I run the tool on an audiobook containing just "first sentence second sentence", and the tool segments the audio into utterances containing "first sentence" and "second sentence", then the transcripts should not be e.g. "first sentence second" and "sentence". Does that make sense?
> > >
> > > Right, the goal of the re-segmentation was to split the long recording into original sentences and make sure that the cut clips contain only the words from the original utterance. The segmentation receives a list of text phrases and its goal is to align these phrases within the long audio. To validate that the re-segmented corpus has the correct alignment, we can analyze the result in the running SDE instance:
> > > 1. Each example contains `text` - the target text utterance we wanted to align within the long audio file. We run segmentation to split the concatenated audio file into small audio clips, each clip corresponding to the original `text`.
> > > 2. `pred_text` - an ASR model transcript of the recovered during the re-segmentation audio clip.
> > > 3. We can sort the re-segmented corpus by `WMR`(word match rate), and look at the `text diff` field (shows the difference between target - `text` and segmentation result - `pred_text`). We can go through and listen to samples with the lowest WMR values. The re-segmentation of the LibriSpeech dev set produced valid segments, the errors in the diff are due to ASR prediction errors, not segmentation.
> > > WER 3.82% is the word error rate of the re-segmented corpus using [QuartzNet15x5Base-En model](https://ngc.nvidia.com/catalog/models/nvidia:nemospeechmodels). Misalignment errors in the segmentation result in a larger gap between ASR model predictions and the reference text. CER of the re-segmented corpus is 1.16%. Overall WER and CER of the original LibriSpeech corpus using the same ASR model are 3.78% and 1.14% correspondingly. The slight difference in score between the original and the re-segmented corpus can be explained by different model’s normalization coefficients due to slightly different segments (as the segmentation dropped some non-speech segments). We manually inspected the examples with the highest WER (CER) values and the corresponding segmented audio clips sound correct.
> > >
> > >
> > > Kind regards,
> > >
> > > Paper246 Authors

---

> ### Comment · Reviewer_SWAm · 2021-10-03
> **Updating my score**
>
> I'd like to increase my score to 6 after the authors' rebuttal---the improved links to code and notebooks, the cleaned-up figures, and the additional LibriSpeech experiment make this a much stronger paper.
>
> (I'm trying to edit my original review, but it doesn't seem to be possible.)

---

### Official Review · Reviewer_w3Bo · 2021-09-20
**A solid toolbox for creating and analyzing speech datasets.**

**Rating:** 8
**Confidence:** 3
**Correctness:** The paper does a great job of demonst…
**Clarity:** This paper is clear and easy to follow.

**Strengths:**

1. A first-of-its-kind toolbox for speech dataset creation and analysis.
2. Practical improvements of the CTC-Segmentation algorithm (processing multiple files in parallel, reducing WER through K-fold re-alignment, etc..)
3. An analysis method that allows to easily find and fix errors in speech datasets.
4. A Russian speech dataset was created from scratch despite the use of a weak ASR model and potential differences between the utterances in the text and those in the audio recordings.


**Weaknesses:**

The dataset created using the CTC-segmentation tool is fairly limited. The reasons behind these limitations are well explained and they showcase the tool’s robustness to potential data issues. However, It would be interesting to see how well the dataset creation method works with a strong ASR model and more suitable raw audio and text data.

**Additional Feedback:**

The paper mentions the importance of speaker variance in speech datasets but I wonder if the data explorer tool allows for speaker-based filtering. Accessing the different statistics mentioned in the paper on a per-speaker basis might be useful in bias studies.

**Documentation:**

The dataset creation and analysis processes are both well documented.

**Ethics:**

I believe that there is no ethical issue in the paper. The authors warn against nonconsensual replication of the voices in speech datasets for TTS models.

**Relation To Prior Work:**

The paper introduces a first-of-its-kind toolbox but the work seems to be clearly positioned in relation to prior work on speech datasets.

**Summary And Contributions:**

This paper proposes a useful open-sourced framework for creating and analyzing speech datasets.

For the purpose of dataset creation, the paper extends an existing CTC-segmentation algorithm into an end-to-end audio-text alignment tool. The tool takes as input long raw audio files and corresponding text utterances and returns a set of processed short text-audio pairs that can be used to train an ASR model.

For the analysis component, the paper proposes an interactive application with practical features such as ASR-based transcription, data exploration, plotting, and error analysis.

Additionally, the paper presents concrete use-cases for both tools by creating a Russian ASR dataset from scratch and providing an analysis of 2 existing datasets.

The contributions made by the authors are clearly demonstrated and the tools presented should be helpful to speech researchers.

---

> ### Author Response · Authors · 2021-09-26
> **Re: A solid toolbox for creating and analyzing speech datasets.**
>
> Dear Reviewer,
>
> Thank you for your feedback and questions.
>
> > Blockquote ... It would be interesting to see how well the dataset creation method works with a strong ASR model and more suitable raw audio and text data.
>
> We’ve applied the same tools (CTC Segmentation and SDE) to create and publicly release a LibriVox-based English dataset for training text-to-speech models (Hi-Fi TTS dataset [3], http://www.openslr.org/109/), a strong pre-trained English ASR model definitely helped a lot. Additionally, we followed the suggestion of Reviewer SWAm and re-segmented the LibriSpeech development set to confirm the sustainability of the toolkit. Please see https://docs.nvidia.com/deeplearning/nemo/user-guide/docs/en/main/tools/speech_data_explorer.html#sde-demo-instance for more details.
>
> > Blockquote The paper mentions the importance of speaker variance in speech datasets but I wonder if the data explorer tool allows for speaker-based filtering. Accessing the different statistics mentioned in the paper on a per-speaker basis might be useful in bias studies.
>
> Currently, SDE does not have speaker-specific visualizations. If the dataset includes speaker information as attributes in the JSON manifest file, then SDE allows filtering and sorting based on these attributes. Adding speaker-based visualizations and analysis is definitely on our roadmap.
>
> Kind regards,
>
> Paper246 Authors

---

> > ### Comment · Reviewer_w3Bo · 2021-09-29
> > **Response**
> >
> > > We’ve applied the same tools (CTC Segmentation and SDE) to create and publicly release a LibriVox-based English dataset for training text-to-speech models (Hi-Fi TTS dataset [3], http://www.openslr.org/109/), a strong pre-trained English ASR model definitely helped a lot. Additionally, we followed the suggestion of Reviewer SWAm and re-segmented the LibriSpeech development set to confirm the sustainability of the toolkit. Please see https://docs.nvidia.com/deeplearning/nemo/user-guide/docs/en/main/tools/speech_data_explorer.html#sde-demo-instance for more details.
> >
> > Adding the LibriSpeech re-segmentation results resolves this concern of mine. Thank you for the nice work.

---

### Decision · Program_Chairs · 2021-10-09

**Decision:**

Accept

**Comment:**

The paper extends the existing NeMO toolkit with tools to perform audio alignment and the production of short units of paired, aligned speech and text. The extensions perform pre- and post-processing on the input audio and text. A tool is also provided to explore a speech dataset by providing visualizations of useful information about its contents, e.g., the length of included transcripts; multi-step realignment is proposed as a mechanism for improving the overall segmentation results. A Russian dataset is constructed using the proposed toolkit to demonstrate its utility. The motivation for the work is clear, as the process of building a speech-based dataset is generally burdensome. For users of NeMO, the additional tools provided are likely to improve the overall process. Generally, reviewers agree that the dataset visualization tool (SDE) has value and may be of use to those creating or exploring datasets. Useful heuristics for dataset creation are proposed as best practices, and the paper is well-written overall.

Some of reviewers' initial concerns were addressed, and those clarifications and imporvements should be in the final version of the paper. It's not clear how automated or turnkey this toolkit is, and which steps need to be managed by hand. In terms of the overall alignments, WER is not as easy to interpret as ground truth segmentations would be. It's also not obvious how applicable the developed tools will be to anyone who is not using NeMO, and the theoretical contribution is strongly influenced by pre-existing CTC models.